# Type-I myosins promote actin polymerization to drive membrane bending in endocytosis

Hetty E Manenschijn[1,2], Andrea Picco[1,3], Markus Mund[1,2], Anne-Sophie Rivier-Cordey[1], Jonas Ries[2], Marko Kaksonen[1,3]*

[1]Department of Biochemistry, University of Geneva, Geneva, Switzerland; [2]Cell Biology and Biophysics Unit, European Molecular Biology Laboratory (EMBL), Heidelberg, Germany; [3]NCCR Chemical Biology, University of Geneva, Geneva, Switzerland

**Abstract** Clathrin-mediated endocytosis in budding yeast requires the formation of a dynamic actin network that produces the force to invaginate the plasma membrane against the intracellular turgor pressure. The type-I myosins Myo3 and Myo5 are important for endocytic membrane reshaping, but mechanistic details of their function remain scarce. Here, we studied the function of Myo3 and Myo5 during endocytosis using quantitative live-cell imaging and genetic perturbations. We show that the type-I myosins promote, in a dose-dependent way, the growth and expansion of the actin network, which controls the speed of membrane and coat internalization. We found that this myosin-activity is independent of the actin nucleation promoting activity of myosins, and cannot be compensated for by increasing actin nucleation. Our results suggest a new mechanism for type-I myosins to produce force by promoting actin filament polymerization.
DOI: https://doi.org/10.7554/eLife.44215.001

*For correspondence:
marko.kaksonen@unige.ch

**Competing interests:** The authors declare that no competing interests exist.

## Introduction

Clathrin-mediated endocytosis is a highly complex process in which the eukaryotic cell deforms its plasma membrane and pinches off a small vesicle (*Kaksonen and Roux, 2018*; *Kirchhausen et al., 2014*; *McMahon and Boucrot, 2011*). Endocytosis requires the dynamic recruitment and disassembly of dozens of different proteins in a highly stereotypical sequence to coordinate cargo recruitment, membrane reshaping, scission and vesicle uncoating (*Kaksonen et al., 2005*). In budding yeast (*Saccharomyces cerevisiae*) and fission yeast (*Schizosaccharomyces pombe*), endocytic membrane bending is strictly dependent on the assembly of a dynamic, branched actin network (*Gachet and Hyams, 2005*; *Kaksonen et al., 2003*), which is thought to provide the force to bend the membrane against the high turgor pressure of yeasts. In metazoan cells, actin polymerization is required for clathrin-mediated endocytosis under certain conditions where the membrane resists bending, for example in case of high membrane tension or strong substrate adhesion (*Batchelder and Yarar, 2010*; *Boulant et al., 2011*).

The endocytic actin structure in yeast consists of thousands of individual protein molecules, which transiently co-assemble to form a highly dynamic, branched actin network (*Goode et al., 2015*). Actin filaments are nucleated through activation of the Arp2/3-complex, which induces the formation of new daughter filaments branching at a 70° angle from a mother filament (*Amann and Pollard, 2001a*; *Amann and Pollard, 2001b*). The main Arp2/3-activating nucleation promoting factors (NPFs) of budding yeast, Las17 (homolog of Wiskott-Aldrich syndrome protein (WASp)) and the type-I myosins Myo3 and Myo5, are recruited to a ring-shaped area on the plasma membrane surrounding the forming invagination, resulting in an inward flowing actin network (*Idrissi et al., 2008*;

*Mund et al., 2018*; *Picco et al., 2015*). This actin network is connected to the tip of the invaginating membrane by the epsins and Sla2, proteins that bind both lipids and actin filaments (*Skruzny et al., 2015*; *Skruzny et al., 2012*). Though it is generally accepted that a large fraction of the force that is required to bend the membrane is provided by actin polymerization via a Brownian ratchet-type mechanism (*Mogilner and Oster, 1996*; *Peskin et al., 1993*), several modeling studies suggest that this force alone is not sufficient to counter the high turgor pressure of the cell (*Carlsson and Bayly, 2014*; *Dmitrieff and Nédélec, 2015*). Additional factors that may contribute to membrane bending include myosin motor activity, membrane curvature generation or stabilization by proteins that bind to the membrane, or local decreases in turgor pressure (*Carlsson, 2018*).

Type-I myosins play an important but enigmatic role in endocytosis. Type-I myosins are monomeric, actin-based motors that use the energy from ATP hydrolysis to power work along actin filaments. Besides an N-terminal motor domain, comprising the ATPase site and an actin-binding pocket, they contain a neck with one or more binding sites for myosin light chains, and a tail which can interact with lipids and other proteins (*Masters et al., 2017*). Generally type-I myosins are involved in linking actin filaments to membranes in diverse cellular contexts such as cell migration, cellular and organellar morphology and maintenance of cortical and plasma membrane tension (*Almeida et al., 2011*; *Dai et al., 1999*; *Diz-Muñoz et al., 2010*; *Hartman et al., 2011*; *McConnell and Tyska, 2010*; *Nambiar et al., 2009*; *Sokac et al., 2006*), but the molecular details of these interactions remain scarce. For several cell types it has been shown that myosin-I molecules are recruited to endocytic sites concomitantly with actin; examples include the two myosin-I paralogs of budding yeast named Myo3 and Myo5 (*Jonsdottir and Li, 2004*; *Sun et al., 2006*), the fission yeast type-I myosin Myo1 (*Arasada et al., 2018*; *Basu et al., 2014*; *Sirotkin et al., 2005*), and Myo1E in metazoan cells (*Cheng et al., 2012*; *Krendel et al., 2007*; *Taylor et al., 2011*). Deletion or depletion of these myosins blocks endocytosis (*Basu et al., 2014*; *Geli and Riezman, 1996*; *Krendel et al., 2007*), but the molecular mechanism by which type-I myosins function remains elusive.

The budding yeasts type-I myosins Myo3 and Myo5 are highly similar and functionally redundant. Deletion of both Myo3 and Myo5 leads to a complete block in endocytosis and a severe growth defect, while single deletions result in no or only very minor defects in endocytosis only under heat stress conditions, and show no growth phenotype (*Anderson et al., 1998*; *Geli and Riezman, 1996*; *Goodson et al., 1996*; *Goodson and Spudich, 1995*). Myo3 and Myo5 share a domain structure consisting of a myosin motor domain, a neck containing two IQ-motifs, and a tail comprising a tail homology 1 (TH1)-domain, an Src homology 3 (SH3)-domain and a central/acidic (CA)-domain (*Goodson and Spudich, 1995*). Myo5 has been shown to have motor activity in vitro (*Sun et al., 2006*), and Myo3 and Myo5 rigor mutants - which bind to actin filaments but cannot release them - block endocytic internalization (*Lechler et al., 2000*; *Lewellyn et al., 2015*; *Sun et al., 2006*). Myo5's TH1-domain has been shown to bind to phospholipids in vitro (*Fernández-Golbano et al., 2014*) and is required for efficient recruitment of Myo5 to endocytic sites (*Grötsch et al., 2010*; *Lewellyn et al., 2015*). The SH3-domain of Myo5 (and presumably Myo3) regulates protein-protein interactions with other endocytic proteins such as Vrp1, Las17, Bbc1 and Pan1 (*Anderson et al., 1998*; *Barker et al., 2007*; *Evangelista et al., 2000*; *Geli et al., 2000*; *Lechler et al., 2000*; *Mochida et al., 2002*; *Sun et al., 2017*). Myo5 SH3-domain interaction with Vrp1 is essential for recruitment of Myo5 to endocytic sites (*Lewellyn et al., 2015*; *Sun et al., 2006*). Finally, the CA-domain together with Vrp1 possess NPF-activity in vitro (*Evangelista et al., 2000*; *Galletta et al., 2012*; *Galletta et al., 2008*; *Geli et al., 2000*; *Lechler et al., 2000*).

While the biochemical properties of Myo3, and Myo5 in particular, have been well characterized in vitro, their functions within the endocytic machinery in vivo remain enigmatic. Myo3 and Myo5 were initially presumed to act mainly as NPFs (*Evangelista et al., 2000*; *Giblin et al., 2011*), however both their CA-domains can be deleted without affecting endocytosis (*Galletta et al., 2008*), indicating that Myo3 and Myo5 have additional functions within the endocytic machinery. Recently it was proposed that Myo3 and Myo5 might dynamically anchor actin filaments to the endocytic site; without this binding the actin filaments would then splay out over the plasma membrane surface, which would interfere with actin network force production (*Evangelista et al., 2000*; *Lewellyn et al., 2015*). Furthermore, myosins could contribute to endocytic force production, for example by pushing actin filaments inwards (*Lechler et al., 2000*; *Sun et al., 2006*). This is in line

with observations that transiently lowering the force requirements can rescue the endocytic defects of Myo1 deletion in fission yeast (*Basu et al., 2014*).

By combining quantitative live-cell imaging with genetic perturbations in budding yeast, we found that the yeast type-I myosins promote actin network growth, and thereby define the speed at which the membrane reshapes. The ability of the myosins to stimulate actin network growth does not depend on their NPF-activity, nor can a deficiency of myosin motors be rescued by increasing actin nucleation, indicating that myosins stimulate actin polymerization rather than nucleation. We propose a model whereby type-I myosins stimulate actin filament polymerization, which contributes to actin network expansion and force production.

## Results

### Deletion of *MYO5* slows down invagination growth and delays scission

Previous studies have shown that deletion of both *MYO3* and *MYO5* results in a complete block in endocytosis (*Geli and Riezman, 1996*; *Goodson et al., 1996*; *Sun et al., 2006*), making it difficult to narrow down how these proteins contribute to each subsequent phase of the endocytic process. Therefore, we decided to quantitatively assess the impact of deleting a single myosin-I gene on the endocytic machinery.

To assess how Myo5 contributes to membrane invagination, we analyzed the dynamics of eight key proteins of the coat (Sla1, Sla2), actin network (Abp1, Cap1, Arc18, Sac6 and Act1) and scission modules (Rvs167) in wild type (WT) and *myo5Δ* cells (*Figure 1A–I*). For each protein, we generated a C-terminally tagged version by integrating EGFP at the genomic locus in a WT or *myo5Δ* background (all proteins in this paper, except for Act1, were tagged at their genomic locus). We performed live-cell imaging and centroid tracking to obtain average trajectories for each protein (*Picco et al., 2015*; *Picco and Kaksonen, 2017*). Additionally we created strains co-expressing Abp1-mCherry, which allows for the precise alignment of the trajectories within each dataset (WT or *myo5Δ*) in respect to the Abp1 trajectory (*Picco et al., 2015*; *Picco and Kaksonen, 2017*). Finally we measured the median number of protein molecules at the endocytic site over time (*Joglekar et al., 2006*; *Picco et al., 2015*). Trajectories were plotted so that for each dataset (WT or *myo5Δ*) the Sla2 trajectory starts at y = 0 (representing the position of the plasma membrane; *Picco et al., 2015*), while the onset of Abp1 assembly is taken as t = 0.

For the coat module, we analyzed two coat proteins Sla1 and Sla2. Sla1 is an abundant component of the endocytic coat and can be used to track the tip of the growing invagination and the newly formed vesicle (*Kaksonen et al., 2003*; *Kukulski et al., 2012a*). In addition, we chose to analyze N-terminally-tagged Sla2, since the fluorophore is located near Sla2's membrane binding domain, meaning it reports the position of the plasma membrane (*Picco et al., 2015*).

Deletion of *MYO5* resulted in a decrease in Sla1 inward movement speed (*Figure 1B*). Note that this defect was present throughout the invagination process, and not limited to any specific phase of the invagination. Sla1 movement speed was reduced from 26.2 ± 1.4 nm/s (mean ± SE) in WT cells to 17.1 ± 1.4 nm/s in *myo5Δ* (p<0.001, Welch 2-sample t-test, see also *Supplementary file 1* table 1). Similar to Sla1, the coat protein Sla2 showed a reduction in inward movement speed (*Figure 1C*). While Sla2's movement speed was reduced in *myo5Δ*, the period during which it moves was prolonged, resulting in a similar end-position for the trajectory. These data suggest that the growth rate of the membrane invagination is reduced in *myo5Δ* cells.

To assess the invagination length at the moment of scission, we followed Rvs167, a BAR-domain-containing protein which assembles around the invagination neck, and whose abrupt disassembly coincides with scission in WT cells (*Kukulski et al., 2012a*; *Picco et al., 2015*). In *myo5Δ* the onset of Rvs167 assembly was delayed relative to the onset of Abp1 assembly (*Figure 1D*, WT: 3.1 s, *myo5Δ*: 5.5 s). However, in both *myo5Δ* and WT cells, Rvs167 was recruited once Sla2 reached an inward depth of ~45 nm (*Figures 1C* and *2A*, WT: 39 ± 6 nm, *myo5Δ*: 47 ± 7 nm, p=0.381). This suggests that Rvs167 recruitment is triggered when the invagination reaches a certain length. In WT cells, the length of the accumulation period of Rvs167 molecules was 3.1 ± 0.1 s (median ± SE, n = 103) while in *myo5Δ* it was extended to 5.5 ± 0.3 s (median ± SE, n = 29), resulting in similar peak amounts of Rvs167 molecules (peak number of molecules and standard error for WT: 136 ± 16, for *myo5Δ*: 110 ± 15, p=0.234). The lengths of accumulation periods for Rvs167 molecules in the two samples

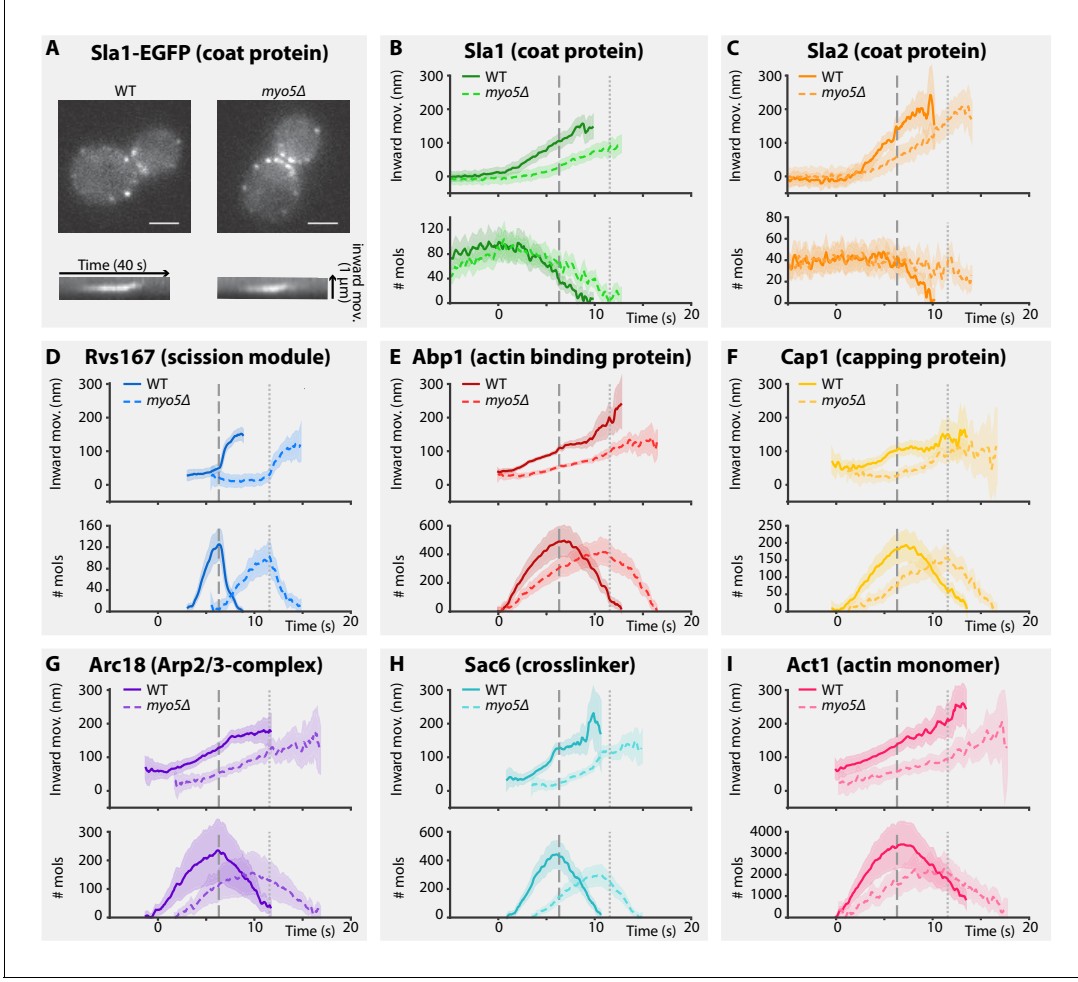

**Figure 1.** Endocytic protein dynamics in WT and *myo5Δ* cells. (**A**) Top: Sla1-EGFP distribution in *WT* and *myo5Δ* cells, bottom: kymographs of Sla1-EGFP patch movement. scalebars: 2 µm. (**B–I**) Inward movement (top graphs) and number of molecules (bottom graphs) of eight different endocytic proteins in WT (darker, solid line) and *myo5Δ* cells (lighter, dashed line). Within each genotype (WT or *myo5Δ*) traces are aligned to each other based on co-alignment with Abp1-mCherry, and plotted such that t = 0 is the onset of Abp1 assembly, and y = 0 is the start of the Sla2 trajectory, marking the position of the plasma membrane. The peak in Rvs167 molecule numbers, representing scission, is marked by dashed vertical lines for the WT dataset, and dotted lines for *myo5Δ*. Shading represents 95% confidence interval.

DOI: https://doi.org/10.7554/eLife.44215.002

The following source data is available for figure 1:

**Source data 1.** The source data of the trajectories plotted in *Figure 1*.

DOI: https://doi.org/10.7554/eLife.44215.003

were different (Mann-Whitney U-test, p value < 0.0001 ). Therefore, Rvs167 molecules accumulated at a slower pace in *myo5Δ* cells, in line with the hypothesis that the invagination growth rate is reduced in *myo5Δ*. At the moment when Rvs167 molecule numbers peaked (indicated by dashed and dotted vertical lines for WT and *myo5Δ* respectively), Sla2 had reached an inward depth of ~150 nm in both WT and *myo5Δ* (*Figures 1C* and *2A*, WT: 145 ± 6 nm, *myo5Δ*: 169 ± 9 nm). Taken together these results indicate that in *myo5Δ* the invagination growth rate is reduced, but since the growth phase is prolonged, similar final invagination lengths are reached.

## Myo5 is required for efficient actin network growth

As myosins are known to interact with actin, we decided to investigate how the actin network is affected by deletion of *MYO5*. We analyzed the dynamics of the actin binding protein Abp1, the capping protein subunit Cap1, the Arp2/3-subunit Arc18, the crosslinker Sac6, and Act1, the actin

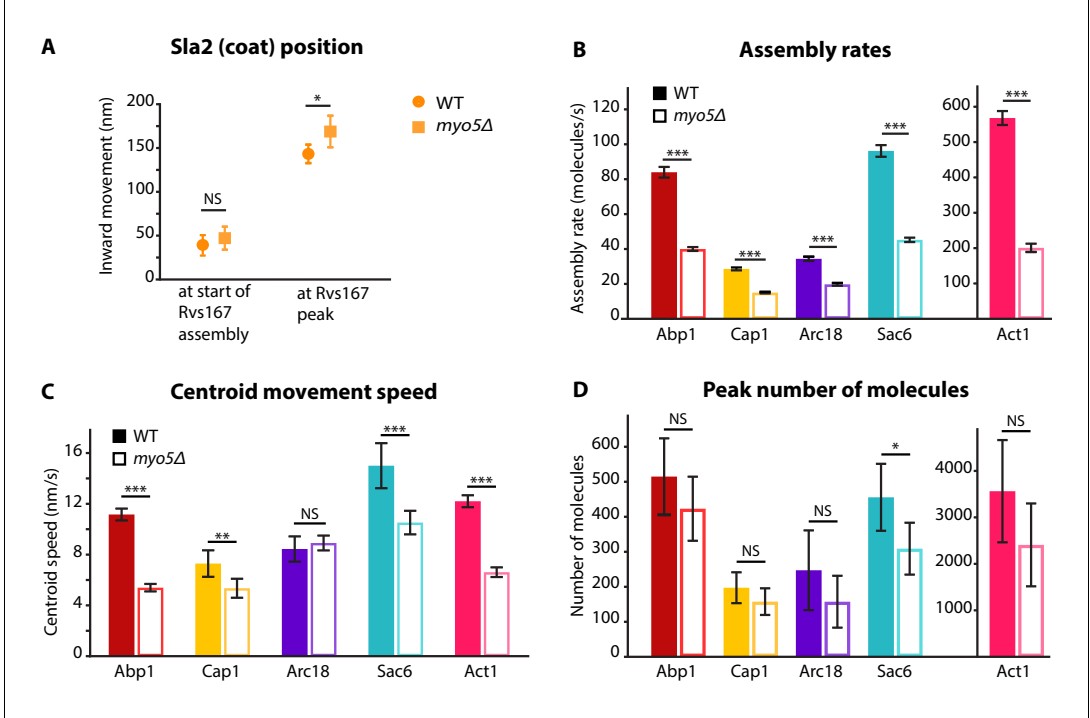

**Figure 2.** Quantification of endocytic protein dynamics in WT and *myo5Δ* cells. (**A**) Sla2 centroid position at different timepoints in WT (circles) and *myo5Δ* (squares). (**B**) Assembly rates for the five different actin network components in WT (darker, solid bars) and *myo5Δ* (lighter, open bars), calculated from trajectories plotted in *Figure 1*. (**C**) Centroid movement speed for the same proteins. (**D**) Peak number of molecules for the same proteins. Error bars represents the 95% confidence interval. WT and *myo5Δ* data were compared using 2-sided z-tests. NS: not significant, *: p≤0.05, **: p≤0.01, ***: p≤0.001.

DOI: https://doi.org/10.7554/eLife.44215.004

The following source data and figure supplement are available for figure 2:

**Source data 1.** Assembly rates, Centroid speeds, and Peak number of molecules represented in *Figure 2*.

DOI: https://doi.org/10.7554/eLife.44215.006

**Figure supplement 1.** phalloidin staining of WT and *myo5Δ* cells.

DOI: https://doi.org/10.7554/eLife.44215.005

monomer, in WT and *myo5Δ* cells (*Figure 1E–I*). As genomic tagging of Act1 is not tolerated (*Wu and Pollard, 2005*), GFP-Act1 was expressed from a plasmid on top of endogenous Act1 expression (*Picco et al., 2015*). Furthermore, Arc18 was tagged with myEGFP, a better tolerated fluorophore by the Arp2/3 complex than EGFP (*Picco et al., 2015*).

We first measured the accumulation rates of these proteins until they reached a peak. We found that the assembly rates for all these proteins were strongly reduced in *myo5Δ* cells as compared to WT cells (*Figure 2B*). However, the durations of the accumulation periods were extended by ~30%, except for Arc18 for which the period was not prolonged. This resulted in comparable peak amounts of the actin network proteins in WT and *myo5Δ* cells (*Figure 2D*), except for Sac6 which was significantly reduced (p=0.018 vs WT).

To check if we could see differences in the actin network using an independent method, we measured filamentous actin amount by quantifying phalloidin-staining intensity in fixed WT and *myo5Δ* cells. We detected a small but significant decrease in the amount of phalloidin per endocytic actin patch in *myo5Δ* cells (12% decrease, p=0.022) (*Figure 2—figure supplement 1*).

The absence of Myo5 also resulted in a decrease in centroid movement speed for Act1, Abp1, Cap1 and Sac6, but not for Arc18 (*Figure 2C*). However, as scission was delayed in *myo5Δ*, the centroids nearly reached the same inward distances at the time of scission as in WT cells.

Taken together these results show that Myo5 stimulates the addition of components to the growing actin network, leading to its expansion.

## Myo3 and Myo5 assemble at the same place and time, but in different amounts

While it is known that Myo5 is recruited to the base of the endocytic invagination at the start of the actin assembly phase (*Idrissi et al., 2008*; *Picco et al., 2015*; *Sun et al., 2006*), Myo3's dynamics have not been quantified. In order to compare Myo3 and Myo5, we performed live-cell imaging and centroid tracking on Myo3-EGFP or Myo5-EGFP together with Abp1-mCherry (*Picco et al., 2015*; *Picco and Kaksonen, 2017*). C-terminal tagging did not detectably affect myosin function as Abp1-mCherry lifetimes were not significantly affected (*Figure 3—figure supplement 1*). Myo3 and Myo5 started to assemble at the endocytic site at the same place and time (*Figure 3A,B*). However, the accumulation rate was higher for Myo5, peaking at about twice as many Myo5 molecules than Myo3 (median number of molecules over the lifetime of the patch and standard error for Myo5: $77 \pm 8$, Myo3: $38 \pm 4$, p<0.001 in 1-sided z-test). The overall similarity in Myo3 and Myo5 recruitment timing and positioning is in line with their reported functional redundancy.

## Myo3 and Myo5 contribute to invagination in a dose-dependent way

Next we assessed whether the deletion of *MYO3* results in the same endocytic defects as the deletion of *MYO5*. We imaged Sla1-EGFP in WT, *myo3Δ* and *myo5Δ* cells, and found that in *myo3Δ* Sla1 dynamics were similar to WT (*Figure 3C*). Because Myo3 and Myo5 are highly similar and are recruited with similar dynamics to the endocytic site, we were surprised to find that deletion of *MYO5*, but not deletion of *MYO3*, had a strong effect on Sla1 inward movement. We wondered if the lack of phenotype in *myo3Δ* was a result of a compensatory increase in Myo5 recruitment, which could mask the effects of the absence of Myo3. We quantified Myo5-EGFP recruitment in *myo3Δ* cells and found it was unaffected by the absence of Myo3 (*Figure 3D*), nor was Myo3 recruitment increased when *MYO5* was deleted. This general lack of feedback may indicate that Myo3 and Myo5 have different binding sites within the endocytic machinery, or that their amounts are not saturating all the binding sites available. The cytoplasmic concentrations of Myo3 and Myo5 have been reported as 152 nM and 172 nM respectively (*Boeke et al., 2014*). We used these values to calculate that on average 34% of the total pool of Myo3 molecules in the cell is located at endocytic sites, compared to 59% of Myo5 molecules (see Materials and methods for calculations). Therefore, Myo5 has a higher affinity than Myo3 for endocytic sites.

In order to see if the Sla1 inward movement rate is determined by the number of myosin molecules at the endocytic site, we created a series of haploid cells carrying combinations of deletions or duplications of the *MYO3* and *MYO5* alleles, and measured both myosin recruitment and Sla1 inward movement in these genetic backgrounds. We also created diploid yeasts carrying deletions of 0, 1, 2, or 3 of the four total myosin-I alleles, in order to obtain endocytic sites containing fewer myosins.

First, we found that the numbers of protein molecules at endocytic sites in WT haploid and WT diploid cells were essentially identical, not only for myosins but also for two other endocytic proteins (*Figure 3D*, *Figure 3—figure supplement 2*). In general, Myo3 and Myo5 recruitment were largely related to their gene dosage, and unaffected by the presence or absence of the other myosin paralog. Exceptions were the *MYO5* duplication strains, where Myo5 recruitment was further increased when *MYO3* was deleted. Also, deletion of a single *MYO5* allele in diploid cells did not significantly reduce Myo5 recruitment, while deletion of a single *MYO3* allele did affect Myo3 recruitment. Taken together, we created a series of strains that have different amounts of myosins at endocytic sites (*Supplementary file 1* table 2).

Intriguingly, we found that Sla1 inward movement speed increased with higher total amounts of Myo3 and Myo5 in a dose-dependent way (*Figure 3E–G*; *Supplementary file 1* table 1). Duplication of the *MYO5* ORF in haploid cells resulted in a slight but significant increase in Sla1 inward movement speed (*Figure 3E*, $30.6 \pm 1.1$ nm/s, p=0.011 vs WT). Sla1 inward movement was similar between WT haploid and WT diploid cells (*Figure 3F*). Furthermore, deletion of all *MYO5* alleles in diploids strongly reduced Sla1 inward movement speeds, phenocopying the *MYO5* deletion in haploid cells, while deletion of one of the two *MYO5* alleles had no effect on Sla1 movement.

To summarize, we found a strong dose-dependent correlation between the number of myosin proteins at the endocytic site, and the Sla1 inward movement rate (*Figure 3G*). This suggests that

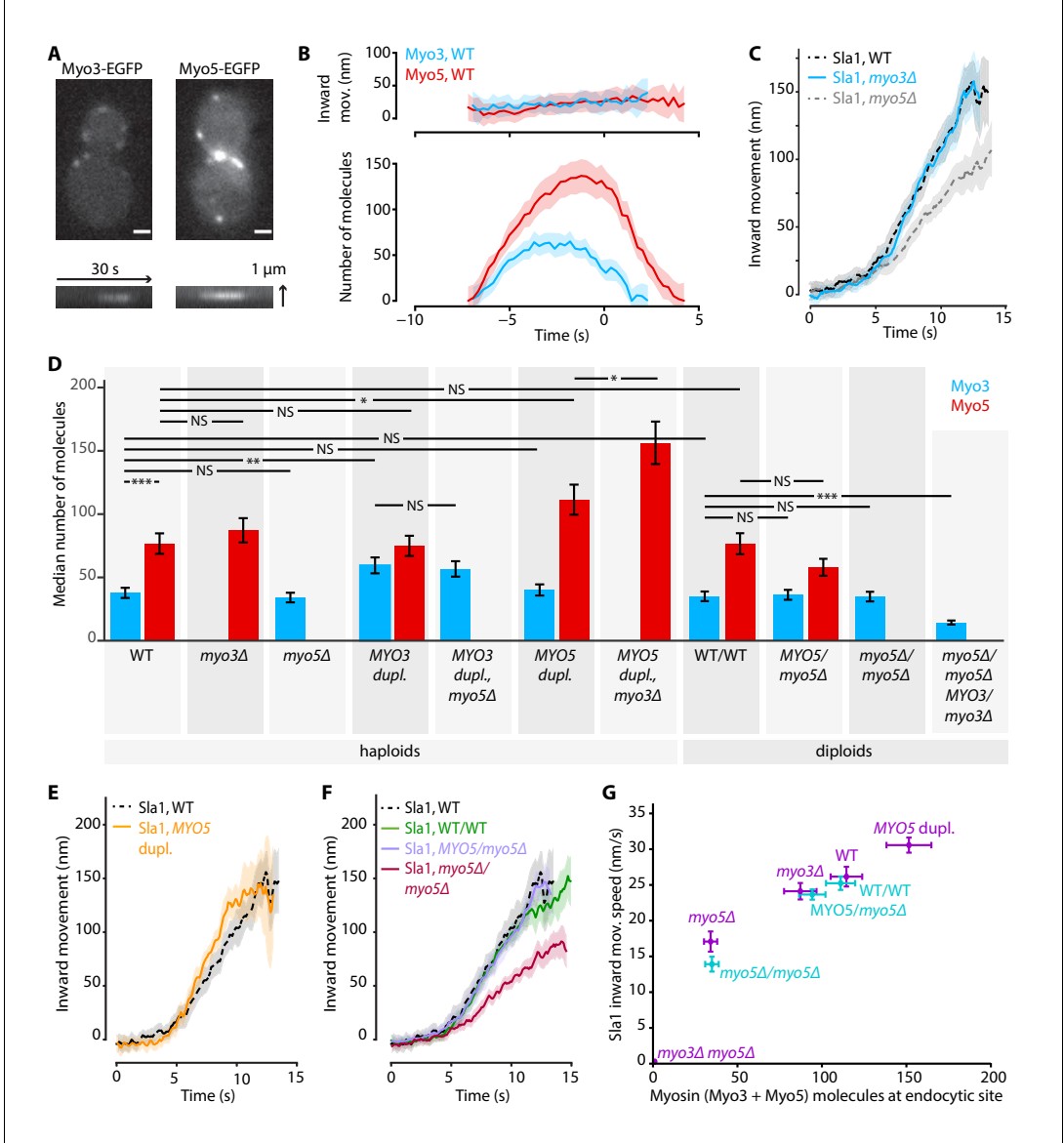

**Figure 3.** Myo3 and Myo5 control Sla1 inward movement rates. (**A**) Top: Myo3-EGFP or Myo5-EGFP patches in WT cells, bottom: kymographs. Scalebar: 1 μm. (**B**) Inward movement (top) and number of molecules (bottom) of Myo3-EGFP and Myo5-EGFP over time at endocytic sites. Traces are aligned in time based on co-alignment with Abp1-mCherry, and plotted such that t = 0 is the peak in Rvs167 molecule numbers (scission). Shading represents 95% confidence interval. (**C**) Average Sla1-EGFP centroid inward movement in WT, *myo3Δ* and *myo5Δ*. Traces are plotted so that the initial position is y = 0, and are aligned in time to the onset of inward movement. Shading represents 95% confidence interval. One of two independent replicates for each genotype is shown. WT and *myo5Δ* trajectories are the same as in **Figure 1**. (**D**) The median number of Myo3-EGFP and/or Myo5-EGFP molecules over the lifetime of an endocytic site are plotted for several haploid and diploid genotypes (gray boxes). Error bars reflect SEM. Molecule numbers were compared using 2-sided z-tests. (**E**) Sla1-EGFP centroid inward movement is plotted over time for WT and *MYO5* duplication cells. (**F**) Sla1-EGFP centroid movement in WT haploids, WT diploids and diploids carrying one or no *MYO5* alleles. (**G**) Sla1 centroid inward movement speed is plotted as a function of the total number of myosins per endocytic site (the sum of the median number of Myo3 and Myo5 molecules). Purple datapoints originate from haploid cells, blue from diploids. The datapoints are labeled with their genotype. Horizontal errorbars are SEM, vertical errorbars SE. mov. = movement, dupl. = duplication, NS = not significant, *: p≤0.05, **: p≤0.01, ***: p≤0.001.

DOI: https://doi.org/10.7554/eLife.44215.007

The following source data and figure supplements are available for figure 3:

**Source data 1.** *Figure 3C, E, and F* average trajectory source data.
DOI: https://doi.org/10.7554/eLife.44215.012

**Figure supplement 1.** C-terminal EGFP-tagging does not affect Abp1-mCherry lifetimes.
DOI: https://doi.org/10.7554/eLife.44215.008

*Figure 3 continued on next page*

Figure 3 continued

**Figure supplement 1—source data 1.** Abp1-mCherry lifetime data in cells expressing no EGFP, Sla1-EGFP, Myo3-EGFP, and Myo5-EGFP.
DOI: https://doi.org/10.7554/eLife.44215.009

**Figure supplement 2.** Endocytic site composition in haploid and diploid yeasts.
DOI: https://doi.org/10.7554/eLife.44215.010

**Figure supplement 2—source data 2.** The median number of molecules represented in *Figure 3—figure supplement 2*.
DOI: https://doi.org/10.7554/eLife.44215.011

myosins control the speed of membrane invagination, potentially via controlling actin network growth.

## The motor activity of Myo5 is necessary for membrane invagination

To test the role of the motor activity in the endocytic invagination process we introduced a glycine 132 to arginine point mutation into the *MYO5* locus. This mutation (Myo5-G132R) is a rigor mutant, which has been reported to inhibit endocytosis and has been suggested to prevent membrane invagination (*Sun et al., 2006*; *Lewellyn et al., 2015*). Consistently, we observed that in Myo5-G132R cells, the Sla1-EGFP invagination movement was completely absent (*Figure 4A*). An average of ~40 Myo5-G132R molecules was recruited to endocytic sites (*Figure 4B*), which is significantly less than the number of Myo5 molecules. However, the reduction of the number of Myo5 molecules by about 50% is not enough to explain the observed strong phenotype. These data suggest that the motor activity is critical for the function of myosins in endocytic membrane invagination.

## Myo5's effect on invagination speed is independent of the NPF activity

Deletion of *MYO5* results in a decrease in invagination speed and negatively affects actin network buildup and movement. Myo5 has a C-terminal CA-domain, which has been shown to possess NPF-activity in vitro (*Geli et al., 2000*; *Idrissi et al., 2002*; *Sun et al., 2006*). To test whether the defects in *myo5Δ* can be explained by a decrease in NPF-activity, we created a truncated Myo5 lacking the CA-domain (*myo5-CAΔ*) (*Galletta et al., 2008*; *Sun et al., 2006*) and measured actin network buildup and invagination rates. Intriguingly we found a slight decrease in the amount of Arc18 at endocytic sites (*Figure 5A*, p=0.043 vs WT), similar to *myo5Δ*. However, while in *myo5Δ* the amount of actin (Act1) was strongly reduced, in *myo5-CAΔ* it was unchanged (*Figure 5B*, p=0.647 vs WT, p=0.001 vs *myo5Δ*). Furthermore, Sla1 inward movement was essentially identical to WT, in contrast to the full *MYO5* deletion (*Figure 5C*). This indicates that the defects in actin network growth and invagination speed in the *myo5Δ* strains cannot be attributed to a decrease in NPF-activity.

## Extra actin cannot compensate for a lack of myosin motors

We wondered if the reduction in Sla1 speed in *myo5Δ* could be rescued by artificially increasing actin polymerization at the endocytic site. Bbc1 is an inhibitor of NPF-activity of Las17 in vitro (*Sun et al., 2006*), and deletion of *BBC1* results in increased Las17 recruitment to the endocytic site and excessive actin polymerization (*Kaksonen et al., 2005*; *Picco et al., 2018*). In *bbc1Δ*, Sla1 centroid movement is initially unperturbed, but accelerates around the time of scission (*Picco et al., 2018*).

Combining the deletions of *MYO5* and *BBC1* balanced out their opposing effects on actin nucleation, resulting in Act1 molecule numbers that were close to WT levels (*Figure 6A*). Intriguingly, Sla1 inward movement showed the combined phenotype of both single deletions: Sla1 centroid movement was initially slowed down, resembling *myo5Δ*, but accelerated at later timepoints, like in the *bbc1Δ* strain (*Figure 6B*). So although the additional deletion of *BBC1* could compensate for the actin defect of *myo5Δ*, it was not sufficient to rescue invagination growth, suggesting that Myo5 has additional roles within the endocytic machinery besides the stimulation of actin filament nucleation.

## Ultrastructural analysis reveals subtle changes in endocytic site morphology in *myo5Δ*

As deletion of *MYO5* affects actin network buildup and coat inward movement, we wondered how the overall ultrastructure of both the actin network and the plasma membrane were impacted. We used superresolution microscopy as described in *Mund et al. (2018)* to measure the dimensions of

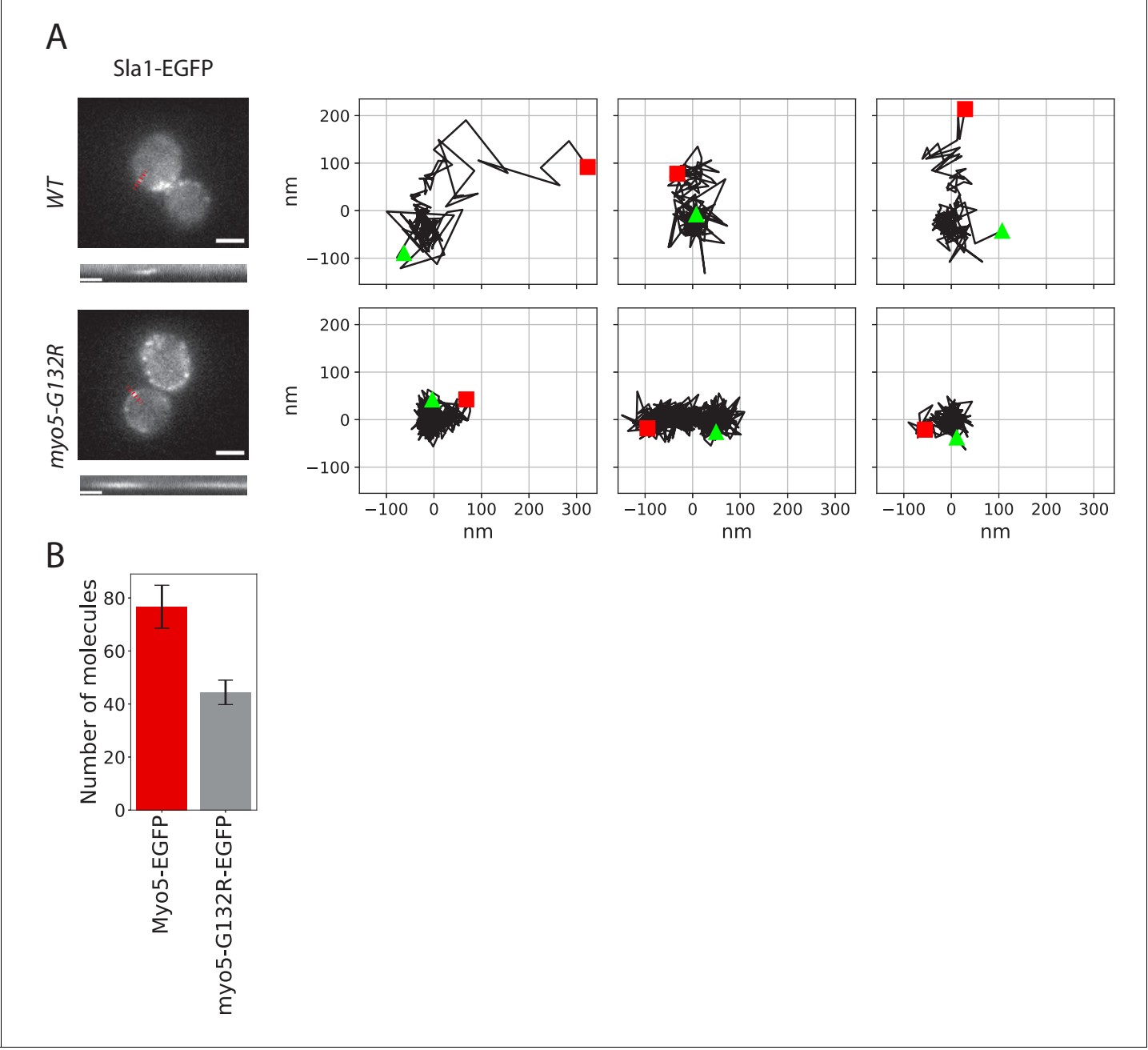

**Figure 4.** Endocytic protein dynamics in *myo5-G132R* cells. (**A**) Left: Sla1-EGFP distribution in WT and *myo5-G132R* cells (scale bars: 2 μm) and kymographs of Sla1-EGFP patch movement (scale bars: 20 s). Right: Examples of Sla1-EGFP trajectories tracked in WT (top row) and *myo5-G132R* cells (bottom row). The trajectories are oriented so that the x-axis is parallel to the cell plasma membrane. The start and the end of each trajectory are marked by a green triangle and a red square respectively. (**B**) The median number of myo5-G132R-EGFP molecules recruited at the endocytic sites plotted aside the median number of Myo5-EGFP molecules as a comparison. Error bars are SEM.

DOI: https://doi.org/10.7554/eLife.44215.013

The following source data is available for figure 4:

**Source data 1.** *Figure 4A* trajectory source data.

DOI: https://doi.org/10.7554/eLife.44215.014

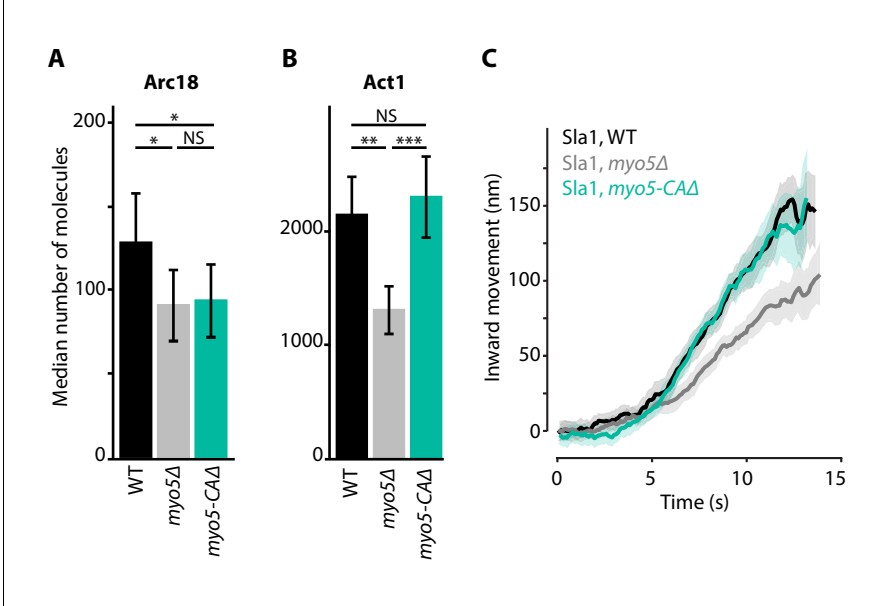

**Figure 5.** Actin network composition and Sla1 dynamics in *myo5-CAΔ*. (**A**) Median number of Arc18 molecules per endocytic site in WT (black), *myo5Δ* (gray) and *myo5-CAΔ* (turquoise) cells. Error bars represent SEM. Molecule numbers were compared using 2-sided z-tests. (**B**) Act1 molecules in the same genotypes. (**C**) Sla1 centroid movement in the same genetic backgrounds (only the inward movement phase is plotted). WT and *myo5Δ* data are the same as in *Figure 1*. Shaded area depicts the 95% confidence interval. NS: not significant, *: p≤0.05, **: p≤0.01, ***: p≤0.001.

DOI: https://doi.org/10.7554/eLife.44215.015

The following source data is available for figure 5:

**Source data 1.** *Figure 5C* average trajectory source data.

DOI: https://doi.org/10.7554/eLife.44215.016

**Source data 2.** The median number of molecules represented in *Figure 5A and B*.

DOI: https://doi.org/10.7554/eLife.44215.017

the actin network in WT and *myo5Δ* cells. For this, we determined the radial density profiles of Abp1, Cap1 and Arc18, and found that the overall protein distributions are very similar (*Figure 7*). However, deletion of *MYO5* resulted in a slight but significant reduction in the outer radii of these proteins, indicating that the actin network on average was narrower.

We were interested in how the defects in actin network assembly and coat motility seen in *myo5Δ* impacted the membrane reshaping during endocytosis. We used correlated light and electron microscopy (CLEM) to investigate membrane shapes in *myo5Δ* cells expressing Sla1-EGFP and Abp1-mCherry, using the protocol described in *Kukulski et al. (2012a)*; *Kukulski et al. (2012b)*. The *myo5Δ* dataset contained both invaginations and vesicles, indicating that productive scission had taken place (*Figure 8*). Intriguingly, the invaginations had larger tip diameters than were reported for WT cells (p=0.003, *Figure 8C*; *Kukulski et al., 2012a*). This increase did not correlate with invagination length (*Figure 8—figure supplement 1*), indicating that the expansion of the invagination tip is present throughout the whole invagination process. Overall, these results indicate that myosin activity impacts the actin network architecture and endocytic membrane morphology.

## Discussion

### Myo3 and Myo5 recruitment

Several binding partners have been shown to participate in the recruitment of Myo3 and Myo5 to the endocytic site, including Vrp1 and actin. However, it was unknown whether the two myosins themselves influence each other's recruitment. We found that Myo3 and Myo5 are recruited to the

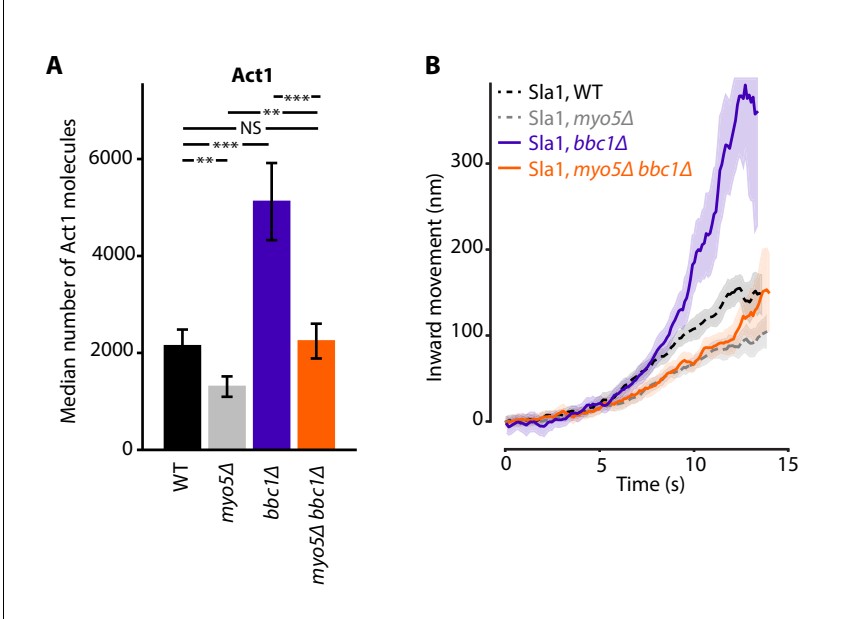

**Figure 6.** Actin content and Sla1 dynamics in *myo5Δ bbc1Δ* double deletion. (**A**) Median number of Act1 molecules per endocytic site in WT, *myo5Δ*, *bbc1Δ* and *myo5Δ bbc1Δ* cells. Error bars are SEM. Values were compared using 2-sided z-tests. WT and *myo5Δ* data are the same as in *Figure 1*. (**B**) Centroid movement of Sla1-EGFP in the same genetic backgrounds. Only the inward movement phase is plotted. WT and *myo5Δ* trajectories are the same as in *Figure 2*. Shaded area represents 95% confidence interval. NS: not significant, *: p≤0.05, **: p≤0.01, ***: p≤0.001.

DOI: https://doi.org/10.7554/eLife.44215.018

The following source data is available for figure 6:

**Source data 1.** *Figure 6B* average trajectory source data.
DOI: https://doi.org/10.7554/eLife.44215.019

**Source data 2.** The median number of Act1 molecules represented in *Figure 6A*.
DOI: https://doi.org/10.7554/eLife.44215.020

endocytic site at the same time and location, but in different amounts (*Figure 3*). Deletion or overexpression of one myosin paralog generally did not affect the recruitment of the other myosin paralog, suggesting that at these expression levels Myo3 and Myo5 do not compete for binding sites. The exception was the *MYO5* gene duplication, where deletion of the *MYO3* allele further increased Myo5 recruitment. Intriguingly, the total number of recruited myosin molecules (Myo3 + Myo5) in these two *MYO5*-duplication strains was the same at ~150 myosins, which may reflect the total number of available myosin binding sites within the endocytic machinery.

## Myo3 and Myo5 regulate actin network growth and invagination speed

Myo3 and Myo5 interact with the endocytic actin network, which is thought to produce the majority of the force that drives membrane bending. We found that deletion of *MYO5* strongly decreased the rate of recruitment for all five actin network proteins tested (*Figures 1* and *2*). This results in a slower inward expansion of the actin network. Although the rate by which new actin components are added is strongly reduced, the peak number of actin network components is generally not affected, as the length of the accumulation phase is extended.

The inward movement rate of coat proteins, which occupy the tip of the invagination and couple it to the expanding actin network, also depends on the amount of myosin-I present at endocytic sites (*Figures 1*, *2* and *3*). We found a strong positive correlation between the total amount of myosins at the endocytic site and coat motility (*Figure 3G*). Note that all haploid data points, except the double deletion, lie on a straight line, implicating that the difference in Sla1 inward movement rate in *myo3Δ* and *myo5Δ* can be explained by the difference in recruitment levels of Myo3 and Myo5. This

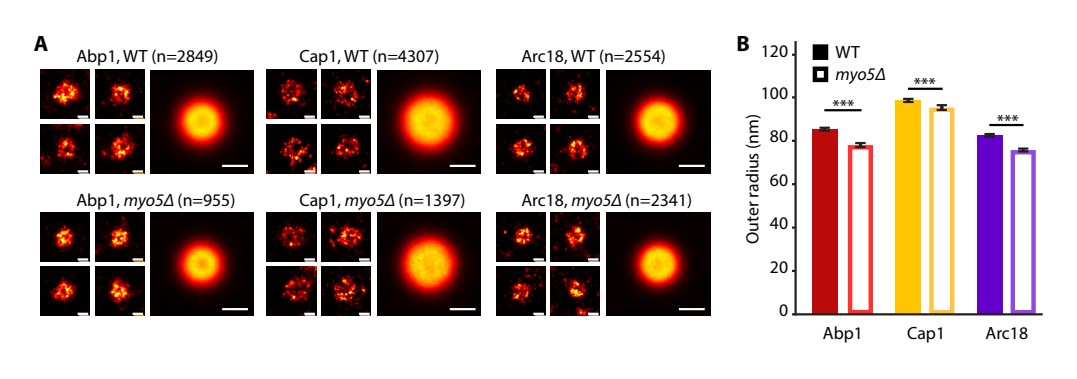

**Figure 7.** The endocytic network is slightly narrower in *myo5Δ*. (**A**) Individual endocytic sites (small pictures) and average protein distributions (big pictures) imaged at the bottom focal plane of the cell for Abp1-mMaple, Cap1-mMaple or Arc18-mMaple in WT (top row) and *myo5Δ* (bottom row) cells. (**B**) Outer radii of the actin network components. Genotypes were compared using 2-sided t-tests. Scalebars: 100 nm. n = number of individual endocytic sites analyzed. ***: p≤0.001. WT data is from *Mund et al. (2018)*.

DOI: https://doi.org/10.7554/eLife.44215.021

The following source data is available for figure 7:

**Source data 1.** The outer radius measurements represented in *Figure 7B*.

DOI: https://doi.org/10.7554/eLife.44215.022

suggests that there is no functional difference between Myo3 and Myo5 under our experimental conditions.

## Temporal coordination in the endocytic machinery

One remarkable feature of the endocytic machinery is the regularity by which endocytic proteins assemble, move and disassemble. Since invagination growth in *myo5Δ* is slower, yet still productive in generating vesicles, we decided to look at how the recruitment and disassembly of different endocytic modules are coordinated. We were especially interested to see if the timing of scission

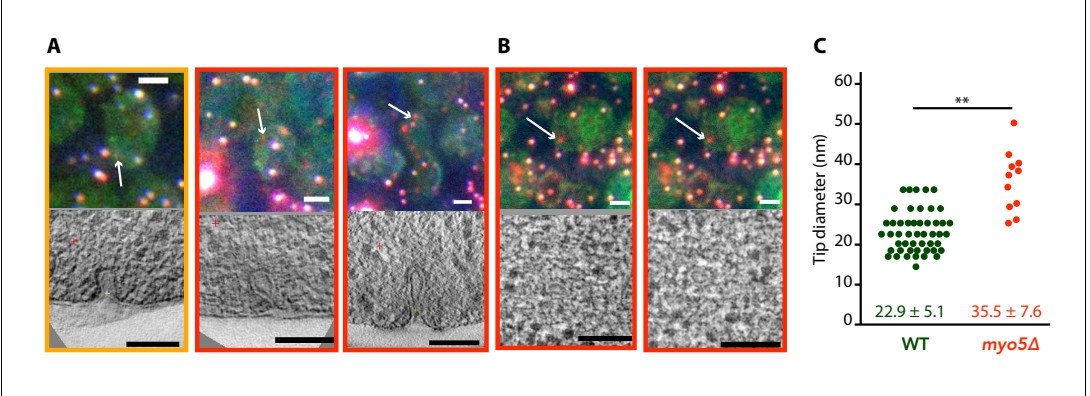

**Figure 8.** Membrane reshaping in *myo5Δ*. (**A**) and (**B**): Gallery of light microscopy (top) and electron tomography (bottom) images of invaginations (**A**) and vesicles (**B**) found in *myo5Δ* Sla1-EGFP Abp1-mCherry cells. The correlated fluorescent spots are indicated by arrows. Note that in (**B**) the same fluorescent spot correlated with two vesicles. The color of the border corresponds to the markers that were present in the endocytic spot (yellow: Sla1-EGFP + Abp1 mCherry, red: only Abp1-mCherry). Scalebars: 2 µm in light microscopy images, 100 nm in electron tomogram slices. (**C**) Diameter of the invagination tip. Only invaginations longer than 40 nm were included. WT data (green) from *Kukulski et al. (2012a)*. **: p≤0.01.

DOI: https://doi.org/10.7554/eLife.44215.023

The following figure supplement is available for figure 8:

**Figure supplement 1.** Endocytic invaginations found in *myo5Δ* cells Invagination tip diameter plotted as a function of invagination length.

DOI: https://doi.org/10.7554/eLife.44215.024

depends on the invagination state of the membrane; or in other words, if scission occurs after a set amount of time or at a set invagination length.

In WT cells, invaginations grow up to ~150 nm long before scission occurs, resulting in the release of a 60 nm long, ellipsoidal vesicle (*Idrissi et al., 2008*; *Kukulski et al., 2012a*). Membrane scission coincides with the disassembly of the BAR-scaffold from the invagination neck (*Picco et al., 2015*), but the causal relationship between these two observations remains unknown. We found that in *myo5Δ* cells the onset of Rvs167 assembly is delayed and subsequently proceeds at a slower pace (*Figure 1D*). This is consistent with a model in which Rvs167 is recruited in a curvature-specific manner; if invagination growth is slowed, the required curvature would be reached later, and further recruitment of Rvs167 would proceed slower. In *myo5Δ* cells, the assembly phase of Rvs167 is extended, resulting in a comparable peak number of Rvs167 molecules. As Rvs167 is thought to scaffold the tubular part of the invagination, reaching a similar amount of Rvs167 proteins would suggest that the invaginations reach the same depth as in WT cells. Furthermore, at the timepoint when Rvs167 molecule numbers peak, the Sla2 patch in *myo5Δ* has traveled a similar distance inward as in WT cells (*Figure 2A*). Finally, the CLEM dataset shows that the distribution of invagination lengths is not affected by the absence of Myo5 (*Figure 8—figure supplement 1*). All in all, we conclude that scission is triggered once the invagination reaches a certain length, rather than after a set period of time. Note that deletion of *RVS167* results in shorter invaginations and smaller vesicles (*Kukulski et al., 2012a*), indicating that scission of shorter invaginations is physically possible.

The absence of Myo5 results in a slowdown of nearly all aspects of endocytic vesicle budding, including actin network assembly and invagination elongation. Intriguingly, the onset of disassembly of the actin and scission modules is also delayed. Therefore, at the crucial moment of scission, in *myo5Δ* most endocytic proteins have reached both similar amounts and positions as they would have in WT cells, suggesting that these modules are interconnected and/or respond to the membrane invagination state. Though in the WT cells the peak of actin assembly coincides almost exactly with the peak in Rvs167 assembly, a slight shift can be observed in the *myo5Δ* cells, suggesting perhaps a defective coupling between the actin and scission modules. Strikingly, the disassembly of the coat does not seem to be tightly connected to scission, as the disassembly of the coat protein Sla1 is linked to the onset of actin assembly or membrane bending rather than scission. At the time of scission in *myo5Δ,* Sla1 disassembly is nearly complete and will only continue for ~1.5 s, compared to ~3.7 s in WT cells. This premature coat disassembly can potentially explain the increased invagination tip diameter we found in our *myo5Δ* CLEM dataset, as fewer coat proteins are present to maintain the diameter of the invagination tip. A similar expansion of invagination tip diameter was also found in the *sac6Δ* strain, and was attributed to the same cause (*Picco et al., 2018*).

## Myosins contribute to force production by stimulating actin polymerization

How do type-I myosins stimulate actin network growth and force production at the endocytic site? The myosin motor activity is critical for their function in endocytosis (*Figure 4*; *Sun et al., 2006*; *Lewellyn et al., 2015*). A recent study proposed that Myo5 stabilizes the actin network by restraining actin filaments to the vicinity of the endocytic site (*Lewellyn et al., 2015*). According to this hypothesis, without myosins the actin network collapses and splays out over the plasma membrane, which would severely interfere with force production. However, our superresolution microscopy data showed the actin network in *myo5Δ* is slightly but significantly narrower (*Figure 7*), which does not support this model.

Both Myo3 and Myo5 carry CA-domains which can activate the Arp2/3-complex and thereby stimulate actin branching. To assess the importance of this NPF-activity in vivo, we created a truncated Myo5 allele lacking this CA-domain. Deletion of the Myo5 CA-domain has no effect on coat movement, nor on Act1 accumulation, in contrast to the full deletion of *MYO5* (*Figure 5*). This is in line with previous studies showing that the deletion of Myo3 and Myo5 CA-domains has no effect on cell growth (*Evangelista et al., 2000*; *Lechler et al., 2000*), endocytic internalization (*Galletta et al., 2008*; *Geli et al., 2000*), or actin patch motility as measured by mean-square-displacement (*Galletta et al., 2008*). Furthermore, artificially increasing actin nucleation by deleting *BBC1* is not sufficient to restore coat inward movement (*Figure 6*). Taken together our results indicate that Myo3 and Myo5 stimulate actin filament polymerization rather than nucleation.

Branched actin networks are thought to produce force via a 'Brownian ratchet' type of mechanism (*Mogilner and Oster, 2003*; *Mogilner and Oster, 1996*; *Peskin et al., 1993*). According to this model, actin filaments tips facing the membrane undergo rapid thermal fluctuations, during which a gap between the tip and the membrane opens up that allows for the addition of a new actin monomer. The elongated filament, which is part of the actin network, creates stresses between the network and the membrane, resulting in membrane deformation. The growth rate of an actin filament is proportional to the probability that an appropriately sized gap opens up for a long enough time to allow the addition of a new monomer, and therefore depends strongly on the load on the filament (*Mogilner and Oster, 2003*; *Mogilner and Oster, 1996*; *Peskin et al., 1993*). The polymerization rate of single actin filaments (*Footer et al., 2007*) and of complex networks (*Bieling et al., 2016*) decreases with increasing loads, until growth completely stalls. In recent years, estimates have been made for both the force required to deform the plasma membrane in yeast (*Dmitrieff and Nédélec, 2015*), and for the amount of force an endocytic actin network could produce (*Berro et al., 2010*; *Carlsson and Bayly, 2014*; *Zhang et al., 2015*), revealing a large gap between the required and produced forces, suggesting that the endocytic ratchet should fail. None of these models have implemented type-I myosins, as it is unclear how and how much they contribute to force production. Based on our results, we propose a working model where type-I myosins increase the polymerization rate of actin filaments by decreasing the load on their tips (*Figure 9*). Myosins could reorient actin filaments facing the plasma membrane, either by pushing them inwards or by rotating them away from the membrane, allowing their further elongation, resulting in actin network growth. In our working model, myosins essentially function as an actin polymerase: they facilitate the addition of new actin monomers to existing filaments under conditions of high loads, thereby allowing actin network growth.

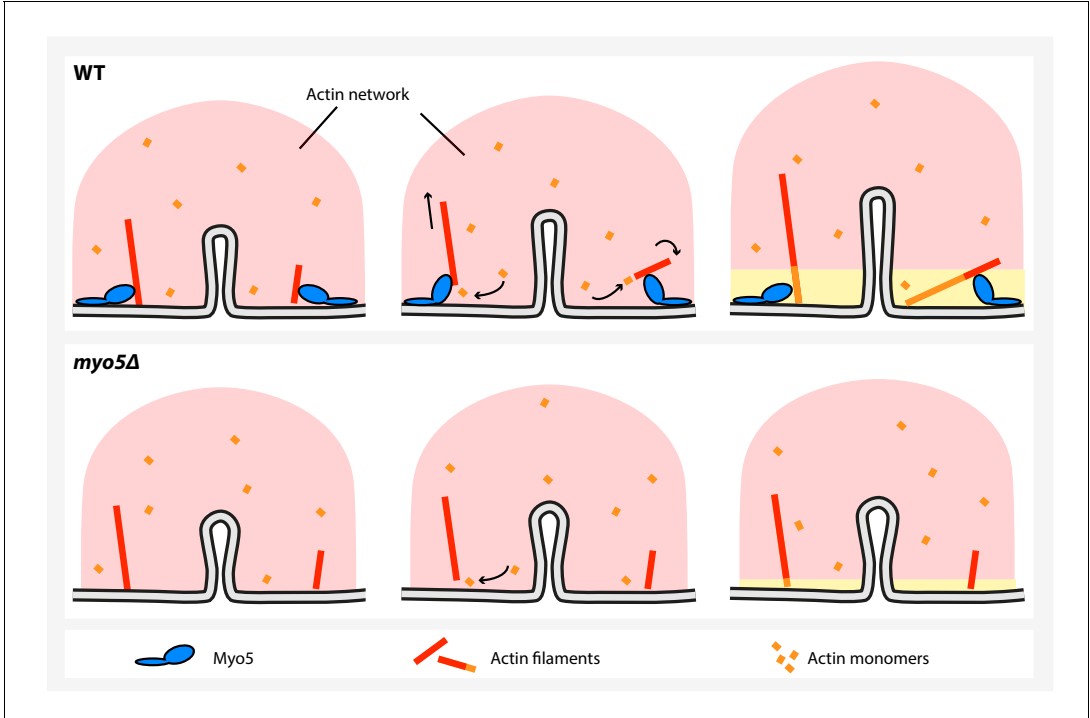

**Figure 9.** A model for myosin-I function during endocytosis in yeast. In WT cells (top), myosin-I reorients or translocates actin filaments whose barbed ends are blocked against the plasma membrane. The barbed ends become accessible to actin monomers again and continue to grow, resulting in actin network expansion. The growing network is coupled to the invagination tip, leading to elongation of the membrane tube. In the *myo5Δ* strain (bottom), many filaments get stuck, decreasing actin polymerization and impairing actin network growth, resulting in slower membrane invagination.
DOI: https://doi.org/10.7554/eLife.44215.025

## Materials and methods

### Strains, media and plasmids

Yeast strains and plasmids are listed in supplementary tables 3 and 4 respectively. Strains were created via homologous recombination with PCR cassettes (*Janke et al., 2004*), and by mating and sporulation. Gene duplications were created according to the protocol described in *Huber et al. (2014)*. EGFP-tagging, gene duplications and truncations were confirmed by sequencing.

### Live cell imaging

Yeast strains were grown to $OD_{600}$ = 0.3–0.8 in SC-TRP or SC-TRP-URA medium. Cells were adhered to ConA-covered coverslips and transferred to the microscope where the CherryTemp spacer and microfluidics chip were added on top. Cells were imaged in SC-TRP or SC-TRP-URA media at 24℃ using the CherryTemp temperature controller. Images were acquired using Olympus IX81 or IX83 wide-field microscopes equipped with 100× 1.45 or 150x 1.45 oil objectives. For single-color imaging, the IX81 and IX83 microscopes were equipped with a single-band emission set GFP-3035c filter (Semrock Brightline) or with the 49002-ET-EGFP (FITC/Cy2, Chroma) set, respectively. We excited the cells with a 488 nm laser line using an 80 ms exposure time and we captured 500-frame long movies using an EMCCD camera ImageEM on the IX81 and ImageEMX2 on the IX83 (Hamamatsu). For dual-color imaging, the IX81 was equipped with a FITC-TRITC dual-band excitation filter set (Olympus) and the and IX83 was equipped with an Olympus BX2/IX3 mounting a ZT488/561rpc and a ZET488/561 m (Chroma). We illuminated the cells for 250 ms using both the 488 and 561 nm laser lines. The emission light was split using a DUAL-view beam splitter (Optical Insights) mounted on the IX81, which we equipped with emission filters 650/75 and 525/50 (Chroma), or with a Gemini (Hamamatsu) mounted on the IX83, which we equipped with emission filters FF03-525/50-25 and FF01-630/92-25 and dichroic Di03-R488/561-t1−25 × 36 (Semrock Brightline). We captured 300-frame long movies using the microscope respective EMCCD cameras. To correct two color images for chromatic aberration we acquired TetraSpecs samples. To quantify the number of molecules, the strain of interest was mixed in a 1:1 ratio with a Nuf2-EGFP strain of the same mating type. We imaged the sample on the IX81 equipped with a U-MGFPHQ (Olympus) filter using a X-CITE 120 PC (EXFO) fluorescent lamp as illumination. We captured 21-frame Z-stacks with 200 nm vertical spacing; each plane was illuminated for 400 ms.

### Image analysis

Single-color and two-color movies were analyzed and average trajectories were obtained as described previously (*Picco et al., 2018*; *Picco et al., 2015*; *Picco and Kaksonen, 2017*); a summary of the number of analyzed trajectories can be found in *Supplementary file 1* table 5. Molecule numbers were calculated as described previously (*Picco et al., 2018*; *Picco et al., 2015*) and an overview can be found in *Supplementary file 1* table 6. For *Figures 1* and *3B*, the average trajectories for each protein within one genotype (either WT or *myo5Δ*) were aligned to each other using the dual-color alignments. The two genotypes were subsequently plotted so that for each genotype y = 0 marks the onset of the Sla2 trajectory, and t = 0 was the start of Abp1 assembly. In the other figures, Sla1 trajectories are aligned in time by eye based on the onset of inward movement. Trajectories were smoothened using a Savitzky-Golay filter and plotted using python or R (*Picco and Kaksonen, 2017*).

To calculate the Sla1 inward movement rates a custom script was used. All Sla1 trajectories were read into R (www.R-project.org). For each trajectory the first timepoint where the patch had moved for ≥50 nm was determined, and a linear regression was fitted through the data starting 1 s before this timepoint and ending 1 s after this timepoint. All individual slopes were pooled, poorly fitted slopes were excluded (st. error of fit ≥8, 0–3 per dataset), and the mean and standard error were calculated. To calculate inward movement rates for the other proteins we fitted a linear regression through the centroid trajectory from the start until the timepoint where the number of molecules peaked. The assembly rates are calculated with a linear regression of the number of molecules over time from the onset of protein assembly until its peak in the number of molecules.

## Correlative light and electron microscopy

Correlated light and electron microscopy was performed as described previously (*Kukulski et al., 2012a*; *Kukulski et al., 2012b*; *Kukulski et al., 2011*). Briefly, MKY3168 cells were grown to $OD_{600}$ = 0.6–1.2 at 24°C, high pressure frozen using an EMPACT2 (Leica) or HPH010 (BAL-TEC) and embedded in Lowycryl HM20 resin (Polysciences, Inc) using an AFS2 (Leica) with robot head for automated pipetting. Using a diamond knife (Drukker), 300 nm thick sections were cut on a microtome (Leica Ultracut) and deposited on carbon-coated copper grids (CF200-Cu, Electron Microscopy Sciences). On the same day, 50 nm TetraSpecks (Life Technologies) were adhered to the sections before fluorescence images were acquired in the red, green and blue channels. Next, 15 nm gold fiducials were adhered to the sections, which was then treated stained with lead citrate. Low-magnification single-axis tilt series (3° increments) and high-magnification single-axis tilt series (1° increments) were acquired over a range from −60° to 60° on a FEI Tecnai F30 operated at 300 kV using serialEM (*Mastronarde, 2005*) and captured on a FEI 4 k Eagle camera or a Gatan OneView 4 k camera. Tomograms were reconstructed using the IMOD package (*Kremer et al., 1996*), and correlations between the light microscopy images and tomograms were performed as described (*Kukulski et al., 2012a*). An overview of the acquired tomograms can be found in *Supplementary file 1* table 7. Invagination sizes were measured using IMOD in a tomographic slice bisecting the invagination.

## Superresolution microscopy

Superresolution imaging and image analysis were performed as described previously (*Mund et al., 2018*; *Mund et al., 2014*). Yeast strains were grown to $OD_{600}$ = 0.6–0.8 in YPAD medium. 1 mL of culture was gently spun down 2.5 min at 500 rcf and 900 uL of medium was removed. The cells were gently resuspended in the remaining medium and allowed to settle onto a ConA coated coverslip for 15 min. The coverslip was then fixed for 15 min in 4 mL of fixation solution (4% formaldehyde, 2% sucrose, PBS). Fixation was stopped by two 15 min incubations in quenching solution (50 mM $NH_4Cl$ in PBS), and coverslips were subsequently washed three times for 5 min in PBS.

The samples were mounted in imaging buffer (50 mM Tris pH 8 in 95% $D_2O$) and then imaged on a custom-built automated superresolution microscope. The output of a laserbox (omicron LightHub) with 405 nm, 488 nm, 561 nm and 640 nm laser lines was despeckled and coupled into a round 105 µm NA 0.22 multimode fiber. The exit of the fiber was then imaged into the sample, which was thereby homogenously illuminated in an area of ~1000 µm² (*Deschamps et al., 2016*). Fluorescence emission was collected by a 160 x NA 1.42 Leica TIRF objective, bandpass filtered using a 600/60 nm filter and focused onto an EMCCD camera (evolve 512, photometrix). During the measurement, the focus was optically stabilized. The microscope was controlled by µManager (*Edelstein et al., 2010*). Before measurements, the back-focal plane image was inspected for air bubbles. Then, using 561 nm illumination of ~10 kW/cm² and an exposure time of 25 ms, movies of typically 20'000–60'000 frames were recorded. The density of localizations was kept constant by automatically adjusting the power of 405 nm illumination.

These movies were analyzed using custom software written in MATLAB. Peaks were detected in the raw images by smoothing, wavelet filtering to remove background, and non-maximum suppression. Peaks above a dynamically adjusted noise threshold were then fitted with a pixelated Gaussian function with a homogenous photon background.

From the localizations, images were reconstructed in the same software. Localizations that were detected in subsequent frames within a 75 nm radius and with a maximum gap of 1 frame were grouped into one. Localizations with a PSF standard deviation higher than 175 nm and a localization precision worse than 30 nm were discarded. An image was reconstructed by overlaying all remaining localizations with a Gaussian with a width proportional to the localization precision. A minimum Gaussian standard deviation of 6 nm was used, and 0.01–0.1% of the brightest pixels were saturated to increase the visibility of the image. Sample drift was corrected by redundant pairwise cross-correlation of ten intermediate images.

Cells and endocytic sites were segmented in the superresolution images. The radial distribution of proteins at the endocytic site was determined by fitting each individual image with a geometric model that describes the structures either as rings or patches (*Mund et al., 2018*). The images were then translationally aligned by the center coordinates obtained from the fit, and the average radial

density was calculated. Sites with outer radii smaller than 30 nm and fewer than 30 localizations were excluded, because these structures were too small to obtain geometric information.

## Calculating the number of myosin molecules in the cytoplasm and at the endocytic sites

We assumed that all Myo3 or Myo5 molecules were either in the cytoplasm or at endocytic sites, and not at other locations in the cell. To calculate the number of Myo (Myo3 or Myo5) molecules in the cytoplasm, we used the following formula

$$Myo_{cyto} = V_{cell} * fraction_{cyto} * [Myo] * N_A$$

in which $Myo_{cyto}$ is the number of myosin molecules in the cytoplasm, $V_{cell}$ is the volume of a haploid yeast cell (42 μm$^3$, BNID 100427, http://bionumbers.hms.harvard.edu), fraction$_{cyto}$ is the cytoplasmic volume fraction (0.29; *Boeke et al., 2014*), [Myo] is the cytosolic concentration of Myo3 or Myo5 (172 nM resp. 152 nM; *Boeke et al., 2014*) and $N_A$ is Avogadro's constant. To calculate the number of Myo molecules at endocytic site, we used the following formulas:

$$E_{Sla1} = p_{Sla1} * a_{cell}$$

$$f = \frac{l_{Myo}}{l_{Sla1}}$$

$$Myo_{endo} = E_{Sla1} * f * N_{Myo}$$

$E_{Sla1}$ is the average number of Sla1 endocytic patches in a yeast cell; it is calculated as the product of $p_{Sla1}$, the number of Sla1 patches per surface area (0.65 μm$^{-1}$; *Brach et al., 2014*), and $a_{Sla1}$, the area of a haploid yeast cell (58.8 μm$^2$, surface area of a 4x4x5 μm prolate sphere). $f$ is the fraction of Sla1 endocytic patches that contain also Myosin molecules; as Myosin and Sla1 lifetimes overlap, it is calculated as the ratio of the two ($l_{Sla1}$ and $l_{Myo}$ are the lifetime of Sla1 and Myosin, respectively). $Myo_{endo}$ is the number of Myosin molecules at the endocytic sites. It is calculated as the average number of Sla1 endocytic patches in yeast cell ($E_{Sla1}$) multiplied by the fraction of those endocytic patches that contain also Myosin molecules ($f$), and $N_{Myo}$, the average number of myosin molecules at an endocytic site, which is calculated as the median number of Myo molecules over the lifetime of a patch (Myo3: 38, Myo5: 77, this paper). We find that an average yeast cell contains 1262 cytoplasmic Myo3 molecules and 654 Myo3 molecules at endocytic sites, giving a total of 1916 Myo3 molecules. Of the total pool of 2732 Myo5 molecules, 1115 are cytoplasmic and 1617 are endocytic.

## Phalloidin staining

Alexa Fluor 488 phalloidin (Mol. Probes, A12379, 300 U) was dissolved in methanol to a final concentration of 6.6 μM. MKY3185 (*myo5Δ*) and MKY0224 (*NUF2-mCherry*) were separately inoculated in 5 mL cultures in SC-TRP O/N at 24°C. These cultures were used to inoculate fresh 5 mL cultures at OD600 = 0.25 and allowed to grow for ~4 hr before being mixed in a 1:1 ratio. 0.5 mL 4% formaldehyde (analytical grade, Fisher Chemical) in PBS was added to 4.5 mL culture mix and incubated for 1.5 hr at 24°C. The fixed cells were washed 3x in PBS and resuspended in 45 uL PBS + 0.1% Triton-X (Roth). 5 uL Alexa488-phalloidin (Alexa Fluor 488 phalloidin, Mol. Probes, A12379, 300 U, dissolved in methanol to a final concentration of 6.6 μM) was added before incubation at RT for 30 min. The cells were washed gently with 500 uL PBS and resuspended in 20 uL PBS. Z-stacks were acquired using the fluorescent lamp in both the GFP and RFP channels, using 200 ms exposure times.

Maximum projections image were created from the Z-stacks and cells were classified based on the presence (WT) or absence (*myo5Δ*) of mCherry signal. Background was reduced by subtracting a 10-pixel median filtered image from the maximum projection. The average fluorescence intensity of actin patches was measured in a 7 × 7 pixel circle.

## Acknowledgements

This work was supported by the European Molecular Biology Laboratory (MM, JR, HM), the European Research Council (ERC CoG-724489, CellStructure to MM, JR) and the Swiss National Science

Foundation (grant 31003A_163267 to MK). We thank the EMBL electron microscopy core facility for support with the CLEM experiments. We thank Tim Wezeman, Karsten Kruse and Daniel Hummel for critical reading of the manuscript.

## Additional information

### Funding

| Funder | Grant reference number | Author |
|---|---|---|
| Swiss National Science Foundation | 31003A_163267 | Marko Kaksonen |
| European Research Council | ERC CoG-724489 | Markus Mund<br>Jonas Ries |

The funders had no role in study design, data collection and interpretation, or the decision to submit the work for publication.

### Author contributions

Hetty E Manenschijn, Conceptualization, Resources, Data curation, Software, Formal analysis, Validation, Investigation, Visualization, Methodology, Writing—original draft, Writing—review and editing; Andrea Picco, Resources, Software, Formal analysis, Validation, Investigation, Methodology, Writing—review and editing; Markus Mund, Resources, Data curation, Formal analysis, Validation, Investigation, Visualization, Methodology, Writing—review and editing; Anne-Sophie Rivier-Cordey, Investigation, Writing—review and editing; Jonas Ries, Conceptualization, Supervision, Funding acquisition, Project administration, Writing—review and editing; Marko Kaksonen, Conceptualization, Supervision, Funding acquisition, Writing—original draft, Project administration, Writing—review and editing

### Author ORCIDs

Hetty E Manenschijn ORCID http://orcid.org/0000-0003-4957-0568
Andrea Picco ORCID https://orcid.org/0000-0003-2548-9183
Markus Mund ORCID https://orcid.org/0000-0001-6449-743X
Jonas Ries ORCID http://orcid.org/0000-0002-6640-9250
Marko Kaksonen ORCID https://orcid.org/0000-0003-3645-7689

### Decision letter and Author response

Decision letter https://doi.org/10.7554/eLife.44215.029
Author response https://doi.org/10.7554/eLife.44215.030

## Additional files

### Supplementary files

• Supplementary file 1. The supplemental tables. Supplemental table 1: Sla1 centroid speeds per genotype. Supplemental table 2: Myosin molecule numbers per genotype. Supplemental table 3: Yeast strains used in this work. Supplemental table 4: Plasmids used in this work. Supplemental table 5: Number of trajectories per dataset. Supplemental table 6: Number of analyzed endocytic sites for quantification of molecule numbers. Supplemental table 7: Summary of CLEM data.
DOI: https://doi.org/10.7554/eLife.44215.026

• Transparent reporting form
DOI: https://doi.org/10.7554/eLife.44215.027

### Data availability

All data generated for this study are included in the manuscript and supporting files.

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
