## [Decision Letter]

Thank you for submitting your article "Type-I myosins promote actin polymerization to drive membrane bending in endocytosis" for consideration by *eLife*. Your article has been reviewed by three peer reviewers, including Christopher G Burd as the Reviewing Editor and Reviewer #1, and the evaluation has been overseen by Anna Akhmanova as the Senior Editor.

The reviewers have discussed the reviews with one another and the Reviewing Editor has drafted this decision to help you prepare a revised submission.

Summary:

The mechanism by which type-I myosins contribute to cellular actin dynamics is unknown. This study provides a detailed description of endocytic defects from quantitative imaging of budding yeast cells with genetic perturbations of the type-I myosins, Myo3 and/or Myo5. The findings indicate that these type-I myosins promote growth of the vesicle-associated actin network by promoting actin filament growth via novel mechanism.

Essential revisions:

For the most part, the reviewers found your study to support the proposed working model of Myo3/5 during endocytosis compelling, though not satisfactorily tested. They feel that a critical test of motor activity is necessary to "actin polymerase" working model. One suggestion to satisfy this would be to examine the effect of point mutations that can reduce (Myo5-S357A) or abolish (Myo5-G132R) the motor domain activity. These mutations were initially published in Lechler et al., 2000 and further studied in the context of endocytosis in budding yeast in Sun et al., 2006. The reviewers feel that this would provide a straightforward test of your model and the results may also substantially raise the significance of your study.

---

## [Author Response]

Essential revisions:For the most part, the reviewers found your study to support the proposed working model of Myo3/5 during endocytosis compelling, though not satisfactorily tested. They feel that a critical test of motor activity is necessary to "actin polymerase" working model. One suggestion to satisfy this would be to examine the effect of point mutations that can reduce (Myo5-S357A) or abolish (Myo5-G132R) the motor domain activity. These mutations were initially published in Lechler et al., 2000 and further studied in the context of endocytosis in budding yeast in Sun et al., 2006. The reviewers feel that this would provide a straightforward test of your model and the results may also substantially raise the significance of your study.

We followed the reviewers’ suggestion and tested the role of Myo5 motor activity by using motor point mutations. We introduced the point mutations G132R or S357A into the *MYO5* locus and tracked the movement of Sla1-EGFP to follow the invagination growth.

In cells expressing Myo5-G132R, the Sla1-EGFP invagination movement was completely halted. This phenotype is described in the new Figure 4A. We could not use our track alignment procedure for this mutant as the invagination movement was completely abolished. The phenotype is demonstrated with kymographs and example x-y centroid trajectories for both the wild type and mutant cells (subsection “The motor activity of Myo5 is necessary for membrane invagination”, subsection “Temporal coordination in the endocytic machinery”, last paragraph, and Figure 4A).

The average number of Myo5-G132R molecules that were recruited to endocytic sites was reduced by approximately half (Figure 4B). However, this ~50% reduction is not enough to explain the block of invagination, as the deletion of on *MYO5* allele in diploid cells causes a minimal phenotype in the invagination speed (see Figure 3G). The Myo5-G132R mutation caused a phenotype that is stronger than the deletion of *MYO5* gene, and comparable to the complete block of endocytosis described for the deletion of both *MYO3* and *MYO5* genes (Figure 3G). This result suggests that myosin motor activity is necessary for the growth of the endocytic invagination and that the G132R mutation acts in a dominant negative fashion. Taken together these findings are in line with the critical importance of the motor domain in promoting invagination growth.

We also tested the effect of the Myo5-S357A mutation, which has been shown to reduce the motor activity of Myo5 in vitro(Sun et al., 2006). Sla1-EGFP movement in Myo5-S357A expressing cells was not significantly different from the movement in cells expressing wild type Myo5 (data not shown). A likely explanation is that the motor activity of Myo5-S357A in vivo is not reduced enough to cause an observable phenotypic difference in these mutant cells. We, therefore, decided not to pursue this mutant further.